# Roles of the interaction with children and families in mediating the association between digital health literacy and well-being of early childhood teachers in Portugal: A cross-sectional study

**Rafaela Rosário**[1,2,3]*, **Sara Barros Araújo**[4], **Ana Silva**[4], **Sílvia Barros**[4]

**1** School of Nursing, University of Minho, Braga, Portugal, **2** Health Sciences Research Unit: Nursing (UICISA: E), Nursing School of Coimbra, Braga, Portugal, **3** Research Centre in Nursing, School of Nursing, University of Minho, Braga, Portugal, **4** Centre for Research and Innovation in Education (inED), Polytechnic Institute of Porto, Porto, Portugal

* rrosario@ese.uminho.pt

**Data Availability Statement:** All DHL descriptive files are available from the https://doi.org/10.34622/datarepositorium/LF7URM.

## Abstract

### Aim

To analyze the associations between early childhood education (ECE) teachers´ digital health literacy (DHL) and well-being, and to determine whether the ECE teachers´ interaction with children and family mediated these associations.

### Methods

A total of 853 early childhood teachers, predominantly women (99.4%) participated in this cross-sectional study. The participants had a mean age of 39.9 years (standard deviation SD = 8.2). Data was collected through an online survey. DHL was assessed using five sub-scales adapted to the new coronavirus context, and the professionals´ well-being was measured using the WHO-5 well-being scale. Binary logistic regression and mediation analysis were used to analyze the data.

### Results

ECE teachers' DHL in dimensions of "information searching" and "determining relevance" had both direct and indirect effect on their well-being. Specifically, higher DHL in these dimensions was associated with better well-being. The dimension "evaluating reliability" had an indirect positive effect on well-being by promoting interaction with children.

### Conclusion

These findings suggest that improving ECE teachers´ DHL could have a positive effect on their well-being and their interactions with children. Therefore, it is recommended to develop health promotion practices aimed at enhancing DHL among ECE teachers. Additionally,

**Funding:** This study was supported by Fundación la Caixa -Social Research Call Covid-19 LL20-3. The funders had no role in study design, data collection and analysis, decision to publish, or preparation of the manuscript.

**Competing interests:** The authors have declared that no competing interests exist.

integrating DHL contents and competencies more prominently into the qualification, further education and training of ECE teachers may help equip them with the necessary skills to access and apply health information effectively. This, in turn, can enhance interactions with children and contribute to their overall well-being.

## Introduction

Health literacy encompasses a range of abilities, motivation and competencies that enable individuals to access, understand, judge and transfer health information into everyday life [1]. It is critical for an individual´s health and well-being [2–7], and has gained increasing attention in research, public health practice and policies [8–10], notably amidst the COVID-19 pandemic and related infodemic [11]. While health literacy has been criticized for its limited focus on individual choices and behaviors [12], contemporary perspectives recognize that individual skills are shaped by the broader environmental demands and complexities they face [13].

The plethora of (mis/des)information and its rapid spread, mainly in the digital realm, has been associated with increased fear among populations and the adoption of harmful health behaviors [14, 15]. During an epidemic, an overwhelming amount of information may be generated, giving rise to what is known as an "infodemic" [11]. Navigating these digital contexts requires new competencies and skills. In response to misinformation, strategies for infodemic management [16] are essential in addressing the challenges faced by communities during health emergencies. This has underscore the importance of developing digital health literacy (DHL) skills [17] that empower individuals to access, understand, and apply health information in their daily lives within the digital age. Consequently, DHL has gained increasing attention from the academic community [18].

Previous research has consistently demonstrated that higher levels of health literacy are linked to improved health outcomes, encompassing various aspect such as better health status [3, 4], healthier behaviors [2] and enhanced subjective well-being [5]. Moreover, a recent systematic review has disclosed that higher health literacy is associated with improved knowledge about blood pressure or hypertension among adults [19], and it significantly improves health-related quality of life in children and adolescents [20]. By investigating the associations between health literacy dimensions and health and well-being in adults, Zhang et al. [21] have found that each dimension exerts both direct and indirect influences on health through various pathways.

The coronavirus disease 2019 (COVID-19) pandemic has disrupted early childhood education (ECE) settings worldwide, leading to closures, online meetings, and physical distance measures. Many countries closed ECE settings in March 2020 and again during the winter in the northern hemisphere [22]. These closures have had profound negative consequences on children [23], families [24], ECE teachers and staff. Additionally, the pandemic has had substantial implications for mental health, with numerous studies highlighting its impact [25, 26]. Since many strategies in ECE settings for promoting well-being involve close physical proximity and touch, ECE teachers are particularly vulnerable in the current circumstances [27]. However, it is worth noting that most of the research conducted thus far has focused on the impact of the pandemic and infodemic on different population groups, rather than specifically in ECE teachers and settings [28, 29]. This knowledge gap underscores the need for comprehensive research that examines the role of DHL in promoting well-being among ECE teachers and its potential impact on their interactions with children and families.

By studying different pathways in healthcare and health literacy, researchers such as Paasche-Orlow and colleagues have proposed that health literacy has an impact on self-care behaviors and lifestyles (e.g., food intake, 24h movement behavior–sleep, sedentary behavior and physical activity) as well as patient-provider relationships [30]. While the role of lifestyles in mediating the associations between health literacy and management of non-communicable diseases has been established [2, 19], there is limited evidence on the specific relationship or interaction between ECE teachers and children and families as an underlying mechanism of DHL. Hence, our study aims to analyse the associations between DHL among ECE teachers and their well-being during the second wave of the pandemic in Portugal immediately before the national lockdown measures were implemented. Additionally, we seek to investigate whether the interactions of ECE teachers with children and families mediate this association, shedding light on the potential mechanisms through which DHL influences their well-being.

## Materials and methods

### Participants

The current study comprised 853 ECE teachers (99.4% of whom females) with a mean of age [mean (standard deviation SD)] of 39.9 (8.2) years old. We contacted all the Portuguese child-care centers from the mainland (18 districts), and autonomous regions of Azores and Madeira. Overall, a total of 2 908 institutions were contacted by e-mail inviting all teachers to participate in the online survey using the platform SurveyMonkey. Data was collected through an online survey from 23rd January to 21st February 2021. Participants completed the informed written consent form before starting the study. The Ethics Commission of the Centre for Research and Innovation in Education (inED) approved the study (number PA8/CE/20). The study implemented appropriate organizational and technical measures to safeguard the collected data and minimize inherent risks associated with data processing. Such measures included pseudonymization, encryption, and access control. The collected data were used for the purposes of the research, and individual results were not disclosed or communicated to any third party.

### Measures

The sociodemographic profile of the ECE teachers was collected. In addition, their well-being was assessed using the 5-item WHO well-being index [31]. This questionnaire asked the teachers to indicate, based on their experience in the past two weeks, how often they felt "cheerful and in good spirits", "active and vigorous", "woke up feeling fresh and rested", and perceived "life as being filled with interesting things". Responses were provided on a 6-point scale, ranging from 0 (at no time) to 5 (all of the time). A composite score was calculated by summing the responses to the 5 items, with higher scores meaning better well-being. Based on the median score, the ECE teachers were divided into two subgroups: those with limited well-being and those with sufficient well-being.

Digital health literacy was assessed using a modified version of the DHL instrument [17], specifically adapted to the new coronavirus context. The instrument consisted of five subscales, each comprising three items that participants were required answer on a 4-point scale (e.g., 1 = very difficult, 4 = very easy). This instrument has been previously used and adapted for use in the Portuguese context [32, 33]. The subscales included the following dimensions: (i) online information searching on coronavirus (reasonable reliability: $\alpha$ = .75); (ii) adding self-generated content (good reliability: $\alpha$ = .81), (iii) evaluating the reliability of coronavirus information (weak reliability: $\alpha$ = .70), (iv) determining personal relevance of coronavirus information (reasonable reliability: $\alpha$ = .72), and (v) protecting privacy on the internet (reliability: $\alpha$ = .36).

Due to the low internal consistency, the fifth dimension, protecting privacy on the internet, was excluded from the subsequent analysis.

ECE teachers´ interaction with children and families was assessed using items derived from pedagogical approaches for 0–3 education and care services [34, 35] and program quality evaluation instruments [36–39]. The assessment encompassed three dimensions: i) the adult-child interactions (11 items); ii) the organization of spaces and materials (8 items), and iii) the interaction with families (5 items). In addition, participants were asked to rate the changes in their pedagogical practices during the pandemic compared to before the pandemic. The items were scored on a 7-item scale, ranging from "much less than before" to "much more frequent than before". Based on the median split, two subgroups were created: one representing ECE teachers with limited interaction and the other representing ECE teachers with sufficient interaction.

### Data analysis

Descriptive statistics were used to explore item-specific normality, and participant characteristics were presented as means, standard deviations (SD) and percentages (%).

Associations between the DHL (exposure variable) and ECE teachers´ well-being (outcome) were analyzed using logistic regression. Adjusted odds ratio (OR) with 95% confidence interval (CI) was considered to determine the strength of association between predictor and outcome. Also, exposure-mediator and mediator-outcome crude associations were tested using logistic regression models (OR and 95% confidence intervals were computed).

Mediation analysis was performed with PROCESS macro 4.0 in IBM SPSS Statistics for Mac, version 27.0 (SPSS Inc. Chicago, IL), with a 0.05 level of significance. The PROCESS macro (model 4) used ECE teachers' well-being as the outcome, DHL dimensions as predictors and ECE teachers-child/family interactions as the mediators variable, with 5.000 bias-corrected bootstrap samples and 95% CI [40]. Since both mediators are significantly correlated we included both mediators jointly in the models [41]. Standardized coefficients and confidence intervals of the estimate were obtained. Total effect and indirect effects are presented.

As potential confounders in all the models, we included any variables hypothesized as affecting DHL, such as the ECE teachers´ age, years of experience, childcare center localization (rural, semi-urban and urban), number of children in the childcare center and, children´s main age. Data analyses were performed using SPSS, version 27.0 (IBM, SPSS Inc. Chicago, IL), considering a level of significance of $< .05$.

### Results

Table 1 depicts the characteristics of the participants. The majority of the ECE teachers were women (99.4%), with a mean (SD) of age of 39.0 (8.2) years old. On average, they had been working in childcare centers for 8.4 (5.8) years. Most ECE teachers had a bachelor's degree (66.5%). The overall score for well-being was 71.4 (18.4). Across the four domains of DHL, the participants perceived "adding self-generated content" and "evaluating reliability" as the most challenging (mean of 3.0, SD of 0.7 and 0.5, respectively). On the other hand, they found "information searching" and "determining relevance" to be the easiest (mean of 3.2 and SD od 0.5 and 0.4, respectively).

After controlling for confounders such as age, years of experience in childcare centers, childcare center location, number of children in the childcare center, children´s age, the results from Table 2 indicate that ECE teachers with higher levels of DHL in the domains of "information searching" and "determining relevance" were more likely to have adequate well-being. Specifically, the odds ratio for adequate well-being was 2.06 (95% CI = 1.41–2.99) for "information searching" and 1.56 (95% CI = 1.05–2.32) for "determining relevance".

**Table 1. The descriptive results of sociodemographic information of ECE teachers and childcare centres.**

|  | N (%) | Missing (n) |
|---|---|---|
| Gender (n (%) women | 848 (99.4) |  |
| **Education level** |  | 4 |
| Bachelor´s degree | 565 (66.5) |  |
| Master´s degree | 282 (33.1) |  |
| Doctorate degree | 4 (0.5) |  |
| **Location of centre** |  |  |
| Urban | 506 (59.3) |  |
| Suburban | 174 (20.4) |  |
| Rural | 173 (20.3) |  |
| **Children's age group** |  |  |
| Less than 12 months | 51 (6.0) |  |
| 12–24 months | 328 (38.8) |  |
| 24–36 months | 365 (43.2) |  |
| Mixed group | 101 (12.0) |  |
|  | Mean (SD |  |
| Age | 39.0 (8.2) |  |
| Number of years working in ECE | 8.4 (5.8) |  |
| Number of children in ECE | 48.1 (32.6) |  |
| Digital Health Literacy |  |  |
| Information searching | 3.2 (0.5) | 296 |
| Adding self-generate content | 3.0 (0.7) | 727 |
| Evaluating reliability | 3.0 (0.5) | 296 |
| Determining relevance | 3.2 (0.4) | 296 |
| ECE teachers Well-being | 71.4 (18. 4) | 127 |
| ECE teachers' Interaction with children | 5.48 (1.08) | 90 |
| ECE teachers' Interaction with families | 5.23 (1.23) | 90 |

Crude associations were tested between DHL (exposure), ECE teachers´ interaction with children and family (mediators), and ECE teachers´ well-being (outcome) (S1 Table), as well as between ECE teachers´ interaction with children and family (mediators) and ECE teachers' well-being (outcome)—S2 Table.

Fig 1 illustrates the direct effect of DHL on ECE teachers´ well-being, mediated by ECE teacher-child interaction (indirect effect). The results indicated that the effect of DHL dimensions on well-being was partially mediated by ECE teacher-child interaction (DHL "information searching": 1.03 (95%CI = 0.21; 2.20); DHL "evaluating reliability": 1.11 (95%CI = 0.29; 2.14); DHL "determining relevance": 1.41 (0.44; 2.76). Even though we found significant direct

**Table 2. Associations between DHL and ECE teachers' well-being.**

|  | DHL Information searching | DHL Adding self generated content | DHL Evaluating reliability | DHL Determining relevance |
|---|---|---|---|---|
|  | OR (95%, CI) | OR (95%, CI) | OR (95%, CI) | OR (95%, CI) |
| ECE teachers´ well-being (median split) | **2.06 (1,41; 2,99)** | 0,86 (0,47; 1,55) | 1,25 (0,89; 1,73) | **1,56 (1.05; 2.32)** |

Model adjusted age, years of experience in childcare centres, geography of the childcare centre, number of children, ages of the children
Bold p<0.05

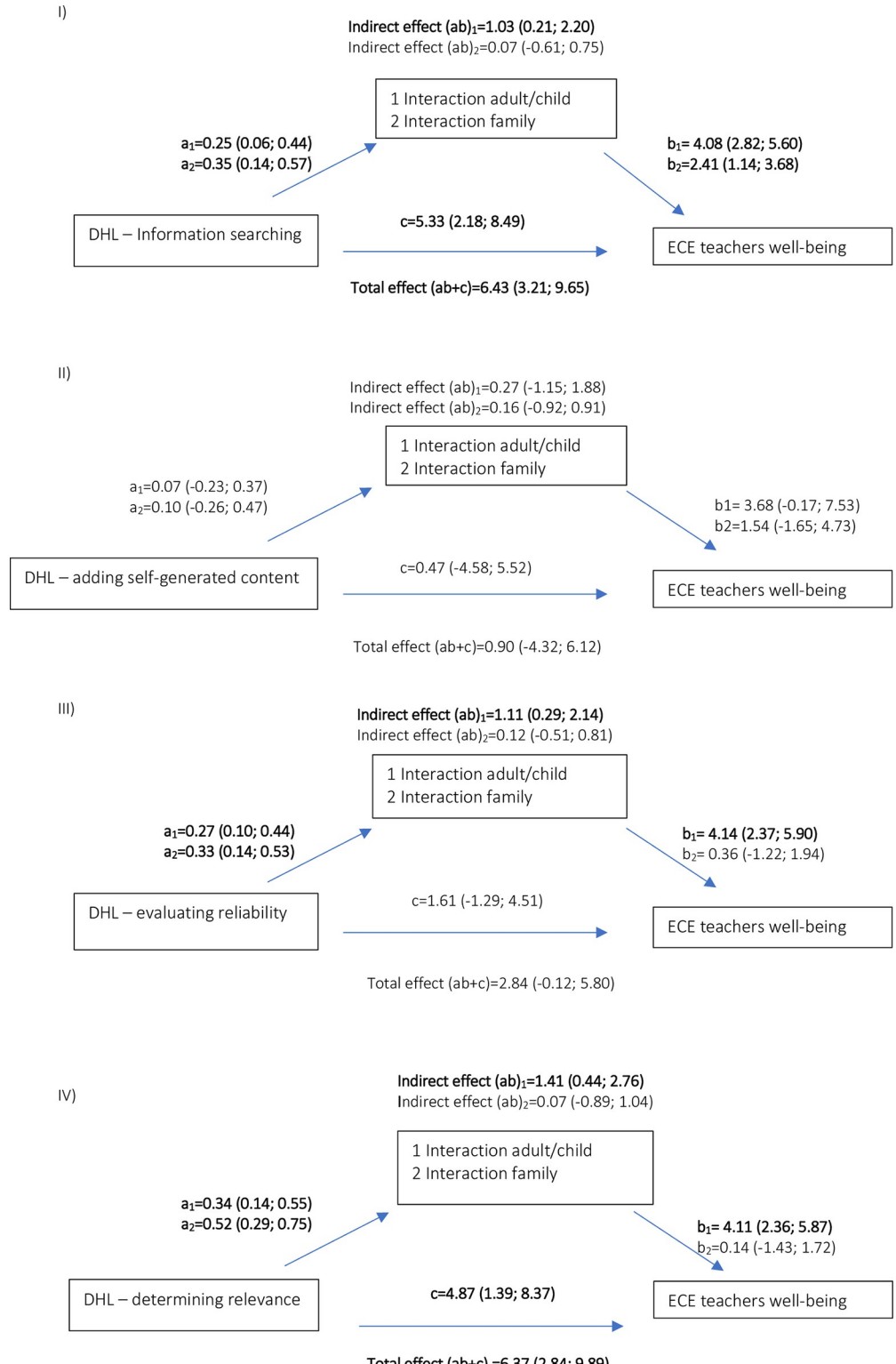

**Fig 1. Direct effect (b and 95% confidence intervals) between DHL dimensions and ECE teachers´well-being.**
Indirect effect (b and 95% confidence intervals) mediated through interaction with children and families. Models were adjusted for age, years of experience in childcare centres, geography of the childcare centre, number of children, ages of the children Bold $p < 0.05$.

associations between DHL dimensions and ECE teachers´ well-being, none of the ECE teachers´ interactions with families appeared to mediate these associations.

## Discussion

The present study revealed significant associations between DHL and well-being among ECE teachers. These findings support the hypothesis that DHL is a determinant of well-being of ECE teachers. Also, we found the aforementioned associations were mediated by ECE teachers-child interaction, particularly in the dimensions of DHL related to "information searching", DHL "evaluating reliability" and DHL "determining relevance".

It is worth noting that most of the ECE teachers assessed themselves as having good to very good skills in the measured items. This is consistent with a previous study conducted with Portuguese university students, where participants also rated themselves as having good to very good skills [32]. This suggests a positive perception of their competencies in health literacy-related tasks among both ECE teachers and university students in Portugal.

The study´s results indicated that DHL dimensions related to "information searching" and "determining relevance" had a direct association with ECE teachers´ well-being. Additionally, these same dimensions, along with DHL "evaluating reliability" had a mediating effect on ECE teachers interaction with children. This suggests that the abilities to search for information and determine its relevance may directly impact the well-being of ECE teachers, while the skill of "evaluating reliability" may modify the aforementioned interactions. The ability of "evaluating reliability" in health information (e.g., decide whether the information is reliable or not, decide whether the information is written with commercial interests, check different websites to see whether they provide the same information) involves interactive health literacy skills, enabling ECE teachers to extract information, derive meaning from different sources, and act independently on new information before making decisions that enhance health and well-being [42]. Interactive health literacy have been associated with various health outcomes [2, 4–7], indicating that individuals who can effectively understand and engage in shared decision-making are more likely to experience better health outcomes. Furthermore, there is a growing recognition of schools as critical settings for improving health literacy [43, 44]. Research has shown that schools play a pivotal role in enhancing health literacy, not only for students but also for teachers and staff. Health literacy education in schools can provide individuals with the necessary knowledge and skills to make informed decisions about their health and well-being [43–45].

As the COVID-19 pandemic progressed, the amount of information available on the topic increased [46]. Therefore, determining which information was most crucial and reliable became challenging (e.g., DHL "evaluating reliability") [47]. This challenge in evaluating the reliability may have hindered direct associations between DHL and the well-being of ECE teachers. It is important to note that vulnerable populations, including children and ECE teachers [27], have often been overlooked when it comes to addressing health literacy [48]. This lack of attention to health literacy, coupled with limited access to health promotion opportunities [49], has potentially restricted better health outcomes for these individuals [50]. To address these limitations and improve the well-being of ECE teachers and other vulnerable populations, there is a need for targeted interventions that promote health literacy and provide equitable access to health promotion initiatives. By enhancing health literacy skills and creating supportive environments, it is possible to empower ECE teachers and other individuals to make informed decisions and improve their overall well-being.

The capacity of "adding self-generated content", which includes skills such as formulating clear health-related questions; expressing opinions or thoughts in writing; and effectively communicating messages, did not show a significant effect on the well-being of ECE teachers. It is

possible that these basic health literacy skills may not have an immediate impact on the interactions between ECE teachers and children/families, or on their overall well-being [42].

The mediating roles of ECE teachers´ interaction with children are consistent with previous studies that have linked health literacy to health outcomes through provider-patient interactions [30]. In the current study the provider-patient interaction is represented by ECE teacher-children interaction and is influenced by health literacy and in turn influences health outcomes (i.e., well-being) [30]. Our findings suggest that among ECE teachers, DHL dimensions, particularly DHL "information searching", DHL "evaluating reliability" and DHL "determining relevance" can enhance their interactions with children and thus promoting ECE teachers´ well-being. However, it is worth noting that the ECE teachers´ interaction with families did not mediate these associations. This could be attributed to the data collection period, which occurred during the first year of the pandemic. This period was marked by several constraints such as ECE services closures and limited active participation of families in the daily life of ECE settings. These circumstances may have hindered the potential mediating effect of ECE teachers' interaction with families on the associations between DHL and well-being.

The current study has several limitations to be acknowledged. First, well-being relied on self-reported perceptions, which could be influenced by ECE teachers' reporting styles and some social desire, limiting the accuracy of capturing well-being. Nevertheless, well-being is a comprehensive construct that encompasses multiple dimensions, including objective and subjective well-being. Future studies could incorporate other dimensions (e.g., physical and built environment, community, and economy) to confirm the current findings. Second, we focused on DHL and thus limiting access to ECE teachers who use the internet less frequently or have weak digital competencies [17]. However, our sample comprises ECE teachers with a minimum of bachelor degree, expected to have intersections with the digital world even in their classrooms [51]. Third, we used cross-sectional data, which preclude the establishment of causal relationships between DHL and well-being. This is, there might be a reciprocal association between them. Actually, a previous study identified health literacy as a clinical risk and as a personal asset [52]. The former reinforces communication as a path to address the needs of low literate individuals, while the latter suggests that health literacy can be developed, as an outcome of health education and communication [52]. Health literacy is developed over time [53], through social interactions in various health contexts [54]. It is expected that people with improvements in health literacy over time, may have more active involvement in health decisions and better health outcomes [50]. In the current study, ECE teachers with higher DHL may have a positive impact on their interactions with children and families, which could in turn lead to better well-being. Still, to further clarify the associations between DHL and health outcomes longitudinal data are required.

This study has also strengths. First, we acknowledge the novelty of the study in its focus on ECE teachers who have been largely overlooked in the COVID-19 pandemic and infodemic. Hence, addressing the idiosyncrasies of the populations contributes for improving health promotion programs and prevention of COVID-19 [55], including health education strategies aiming at strengthening DHL and tackling the infodemic. Second, the overall reliability of each subscale of the of the DHL related with COVID-19 was considered with satisfying Cronbach alpha scores. Finally, we performed the analyses adjusting for important confounders, considered determinants of DHL.

## Conclusion

The present study examined both direct and indirect paths of the DHL dimensions on ECE teachers' well-being- The results indicated that better perceived capacities in information

searching and determining relevance could lead to adequate well-being. Furthermore, improved skills in information searching, evaluating reliability and determining relevance positively impacted ECE teachers interaction with children, which in turn contributed to their well-being. Based on these findings, it is recommended that health promotion programs be developed specifically to enhance DHL among ECE teachers. By improving their DHL, ECE teachers can experience improved well-being and enhance their interaction with children.

## Supporting information

**S1 Table. Associations between DHL and ECE teachers´ well-being and their interaction with children and families.**
(DOCX)

**S2 Table. Associations between ECE teacher´s interaction with children, families and emotional climate, and well-being.**
(DOCX)

## Author Contributions

**Conceptualization:** Rafaela Rosário, Sílvia Barros.

**Formal analysis:** Rafaela Rosário, Ana Silva, Sílvia Barros.

**Funding acquisition:** Rafaela Rosário, Sara Barros Araújo.

**Investigation:** Rafaela Rosário, Sara Barros Araújo, Ana Silva, Sílvia Barros.

**Methodology:** Rafaela Rosário, Sara Barros Araújo, Sílvia Barros.

**Project administration:** Rafaela Rosário, Sara Barros Araújo, Sílvia Barros.

**Resources:** Sara Barros Araújo, Sílvia Barros.

**Validation:** Sílvia Barros.

**Writing – original draft:** Rafaela Rosário.

**Writing – review & editing:** Rafaela Rosário, Sara Barros Araújo, Ana Silva, Sílvia Barros.

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
