## [Decision Letter · Decision Letter 0]

2 Mar 2023

PONE-D-22-23884How different digital health literacy dimensions are associated with early childhood teachers well-being: the mediating role of their interaction with children and familiesPLOS ONE

Dear Dr. Rosário,

Thank you for submitting your manuscript to PLOS ONE. After careful consideration, we feel that it has merit but does not fully meet PLOS ONE’s publication criteria as it currently stands. Therefore, we invite you to submit a revised version of the manuscript that addresses the points raised during the review process.

We look forward to receiving your revised manuscript.

Kind regards,

Ali Garavand

Academic Editor

PLOS ONE

2. Please change "female” or "male" to "woman” or "man" as appropriate, when used as a noun (see for instance https://apastyle.apa.org/style-grammar-guidelines/bias-free-language/gender).

5. Please include your tables as part of your main manuscript and remove the individual files. Please note that supplementary tables (should remain/ be uploaded) as separate "supporting information" files

Reviewers' comments:

Reviewer's Responses to Questions

**Comments to the Author**

1. Is the manuscript technically sound, and do the data support the conclusions?

Reviewer #1: Yes

Reviewer #2: Yes

2. Has the statistical analysis been performed appropriately and rigorously? 

Reviewer #1: Yes

Reviewer #2: Yes

3. Have the authors made all data underlying the findings in their manuscript fully available?

Reviewer #1: Yes

Reviewer #2: Yes

4. Is the manuscript presented in an intelligible fashion and written in standard English?

Reviewer #1: Yes

Reviewer #2: No

5. Review Comments to the Author

Reviewer #1: The author has clearly written the following points to support the manuscript.

1. Sampling technique

2. Data analysis

3. Linkage to conclusion

The author has used binary logistic regression and mediation analysis that suits to the research topic to research desired results. The data has been collected mostly from 853 respondents which has been clearly stated.

The author is requested to read through the manuscript to get proper connectivity between the sub-topics used.

Reviewer #2: That is a good article but I think it needs English edition.

The Introduction is so weak. Please discuss more about necessity, previous study. It is also proposed to present the findings of studies that are closer to the research topic and have pointed out the link between digital health literacy and educational environments.

In the method, ethical considerations should be mentioned with more detail

In the discussion section, please provide more studies related to the research topic and compare the findings of the current research with other related studies.

The strengths and weaknesses of the study should be presented at the end of the discussion.

Some references need to be updated.

6. PLOS authors have the option to publish the peer review history of their article (what does this mean?). If published, this will include your full peer review and any attached files.

Reviewer #1: No

Reviewer #2: No

---

## [Author Response · Author response to Decision Letter 0]

13 Apr 2023

Response to the Editor:

Authors #1

We thank the editor, we updated the style requirements. Please see the revised version of the manuscript.

2-Please change "female” or "male" to "woman” or "man" as appropriate, when used as a noun

Authors #2

We thank the editor, we updated the manuscript according with the suggestion.

3-We note that the grant information you provided in the ‘Funding Information’ and ‘Financial Disclosure’ sections do not match.

Authors #3

We thank the editor, we updated the manuscript according with the suggestion.

4- Data Availability statement.

Authors #4

We thank the editor, all DHL descriptive files are available from the 

https://doi.org/10.34622/datarepositorium/LF7URM

5- Please include your tables as part of your main manuscript and remove the individual files.

Authors #5

We thank the editor, we included the tables and figure into the main manuscript and removed the individual files.

6- Please include captions for your Supporting Information files at the end of your manuscript, and update any in-text citations to match accordingly.

Authors #6

We thank the editor, we included the captions for our Supporting Information files at the end of your manuscript.

7- Please review your reference list to ensure that it is complete and correct.

Authors #7

The references are complete and correct.

Response to the Reviewers:

Reviewer #1:

The author has clearly written the following points to support the manuscript.

1. Sampling technique

2. Data analysis

3. Linkage to conclusion

Authors #1

We thank the reviewer for this reinforcement.

The author has used binary logistic regression and mediation analysis that suits to the research topic to research desired results. The data has been collected mostly from 853 respondents which has been clearly stated.

The author is requested to read through the manuscript to get proper connectivity between the sub-topics used.

Authors #2

We thank the reviewer for addressing this issue. We have included a proper connectivity between the binary logistic regression and mediation analysis. Please see the revised discussion of the manuscript. It now reads:

“Hence, ECE teachers may be better equipped to access and apply health information, enhancing interactions with children, and ultimately improving their well-being.” (lines 37-39.) 

“Figure 1 shows the direct effect of DHL on ECE teachers´ well-being, mediated by ECE teacher-child interaction (indirect effect). Our findings indicate that the effect of DHL dimensions on well-being was partially mediated by ECE teacher-child interaction” (lines 190-193).

Reviewer #2: 

That is a good article but I think it needs English edition.

Authors #1

We thank the reviewer for pointing the need of English edition. We updated the language throughout the manuscript. Please see the revised version of the manuscript.

The Introduction is so weak. Please discuss more about necessity, previous study. It is also proposed to present the findings of studies that are closer to the research topic and have pointed out the link between digital health literacy and educational environments.

Authors #2

We thank the reviewer for pointing this. We discussed more about the need and findings from other studies, please see lines 38-42 and 45-52; it now reads: “Although health literacy has been criticized for neglecting the wider societal and structural dynamics that shape the individuals´ choices (1), contemporary health literacy explanations acknowledge that individuals´ skills are largely influenced by the environmental demands and complexities they face (2)”

“During an epidemic, an overwhelming amount of information may be generated, leading to an "infodemic" (3). To navigate these digital contexts, new competencies are required. In response to misinformation, strategies for infodemic management (4) are essential to address the challenges faced by communities during health emergencies. This has highlighted the importance of developing digital health literacy (DHL) skills (5) to enable individuals to access, understand, and apply health information in their daily lives in the digital age. Consequently, DHL has gained increasing attention from the academic community (6).”

In the method, ethical considerations should be mentioned with more detail

Authors #3

We thank the reviewer for pointing this out. Ethical considerations were detailed, please lines 102-106. It now reads “The study implemented appropriate organizational and technical measures to safeguard the collected data and minimize inherent risks associated with data processing. Such measures included pseudonymization, encryption, and access control. The collected data were used for the purposes of the research, and individual results were not disclosed or communicated to any third party.”

In the discussion section, please provide more studies related to the research topic and compare the findings of the current research with other related studies.

Authors #4

We thank the reviewer for pointing this out. We included more studies related to the research topic and compare the findings of the current research with other related studies. Please see the revised version of the manuscript. It now reads: 

“There is a growing body of literature that has found associations between interactive health literacy and various health outcomes (7-11). This suggests that individuals who are able to effectively understand and engage in shared decision-making are more likely to have better health outcomes. Also, there is a growing body of evidence highlighting schools as critical settings for improving health literacy. (12, 13). Research has shown that schools can play a crucial role in improving health literacy, not only for students but also for teachers and staff. Health literacy education in schools can provide individuals with the knowledge and skills necessary to make informed decisions about their health and well-being (12-14).” (lines 339-347).

The strengths and weaknesses of the study should be presented at the end of the discussion.

Authors #5

We thank the reviewer for pointing this out. The limitations and strengths of the study were presented at the end of the discussion.

Some references need to be updated.

Authors #6

Some references were updated, please see the revised version of the manuscript.

References

1. The Lancet. Why is health literacy failing so many? Lancet (London, England). 2022;400(10364).

2. Nutbeam D, Muscat D. Health Promotion Glossary 2021. Health promotion international. 2021;36(6).

3. Zarocostas J. How to fight an infodemic. Lancet. 2020;395(10225):676.

4. World Health Organization. WHO competency framework: building a response workforce to manage infodemics. Geneva: World Health Organization; 2021.

5. van der Vaart R, Drossaert C. Development of the Digital Health Literacy Instrument: Measuring a Broad Spectrum of Health 1.0 and Health 2.0 Skills. J Med Internet Res. 2017;19(1):e27.

6. Yang K, Hu Y, Qi H. Digital Health Literacy: Bibliometric Analysis. Journal of medical Internet research. 2022;24(7).

7. Pelikan J, Ganahl K, Roethlin F. Health literacy as a determinant, mediator and/or moderator of health: empirical models using the European Health Literacy Survey dataset. Global health promotion. 2018.

8. Brørs G, Dalen H, Allore H, Deaton C, Fridlund B, Norman C, et al. The association of electronic health literacy with behavioural and psychological coronary artery disease risk factors in patients after percutaneous coronary intervention: a 12-month follow-up study. European heart journal Digital health. 2023;4(2).

9. Brørs G, Dalen H, Allore H, Deaton C, Fridlund B, Osborne R, et al. Health Literacy and Risk Factors for Coronary Artery Disease (From the CONCARD PCI Study). The American journal of cardiology. 2022.

10. Wang L, Fang H, Xia Q, Liu X, Chen Y, Zhou P, et al. Health literacy and exercise-focused interventions on clinical measurements in Chinese diabetes patients: A cluster randomized controlled trial. EClinicalMedicine. 2019;17:100211.

11. Hirooka N, Kusano T, Kinoshita S, Aoyagi R, Saito K, Nakamoto H. Association between health literacy and purpose in life and life satisfaction among health management specialists: a cross-sectional study. Scientific reports. 2022;12(1).

12. Paakkari L, Balch-Crystal E, Manu M, Ruotsalainen J, Salminen J, Ulvinen E, et al. Health-literacy education drives empowerment and agency. Lancet (London, England). 2023;401(10374).

13. Okan O, Paakkari L, Jourdan D, Barnekow V, Weber M. The urgent need to address health literacy in schools. Lancet (London, England). 2023;401(10374).

14. Nutbeam D. Health literacy as a public goal: a challenge for contemporary health education and communication strategies into the 21st century. Health Promot Int. 2000;15(3):259-67.

---

## [Decision Letter · Decision Letter 1]

16 Jun 2023

PONE-D-22-23884R1How different digital health literacy dimensions are associated with early childhood teachers well-being: the mediating role of their interaction with children and familiesPLOS ONE

Dear Dr. Rosário

Thank you for submitting your manuscript to PLOS ONE. After careful consideration, we feel that it has merit but does not fully meet PLOS ONE’s publication criteria as it currently stands. Therefore, we invite you to submit a revised version of the manuscript that addresses the points raised during the review process.

We look forward to receiving your revised manuscript.

Kind regards,

Ali Garavand

Academic Editor

PLOS ONE

Journal Requirements:

Reviewers' comments:

Reviewer's Responses to Questions

**Comments to the Author**

1. If the authors have adequately addressed your comments raised in a previous round of review and you feel that this manuscript is now acceptable for publication, you may indicate that here to bypass the “Comments to the Author” section, enter your conflict of interest statement in the “Confidential to Editor” section, and submit your "Accept" recommendation.

Reviewer #2: (No Response)

Reviewer #3: All comments have been addressed

2. Is the manuscript technically sound, and do the data support the conclusions?

Reviewer #2: Yes

Reviewer #3: Yes

3. Has the statistical analysis been performed appropriately and rigorously? 

Reviewer #2: Yes

Reviewer #3: Yes

4. Have the authors made all data underlying the findings in their manuscript fully available?

Reviewer #2: Yes

Reviewer #3: Yes

5. Is the manuscript presented in an intelligible fashion and written in standard English?

Reviewer #2: No

Reviewer #3: Yes

6. Review Comments to the Author

Reviewer #2: I would like to thank you for this research.

In general, the manuscript is written with an easy-to-follow and readable layout, the abstract is good and the problem is well-defined. The Study Area is clearly and appropriately defined. However, the manuscript needs some modifications including:

Please show your study type in the title (Place, time and type of study)

The keywords should be set according to the Mesh term.

Were all the teachers of 2908 institutions invited to the study?

Refer to the table numbers in the text of the article.

A thorough revision of the language is needed to enhance the readability of the text.

Reviewer #3: The authors have addressed satisfactorily the comments from the reviewers. I have also re-read the article, and I think it is ready for acceptance.

7. PLOS authors have the option to publish the peer review history of their article (what does this mean?). If published, this will include your full peer review and any attached files.

Reviewer #2: No

Reviewer #3: No

---

## [Author Response · Author response to Decision Letter 1]

10 Jul 2023

Response to the Reviewer:

Reviewer #2:

I would like to thank you for this research.

In general, the manuscript is written with an easy-to-follow and readable layout, the abstract is good and the problem is well-defined. 

Authors #1

We thank the reviewer for this reinforcement.

The Study Area is clearly and appropriately defined. However, the manuscript needs some modifications including:

Please show your study type in the title (Place, time and type of study)

Authors #2

We appreciate the reviewer's suggestion. The title of the study has been revised to: "Roles of the interaction with children and families in mediating the association between digital health literacy and well-being of early childhood teachers in Portugal: a cross-sectional study."

The keywords should be set according to the Mesh term.

Authors #3

We appreciate the reviewer's feedback. The keywords have been revised to align with MeSH terms. The revised version of the manuscript now includes the following keywords: school teachers, health, health literacy, infant, child, preschool, and family.

Were all the teachers of 2908 institutions invited to the study?

Authors #4

All the teachers from the 2 908 institutions contacted by e-mail, were invited to participate in the online survey using the platform SurveyMonkey. This information was clarified in the methods section. It now reads: “Overall, a total of 2 908 institutions were contacted by e-mail inviting all teachers to participate in the online survey using the platform SurveyMonkey” (lines 300-301)

Refer to the table numbers in the text of the article.

Authors #5

We included the table numbers in the text of the manuscript. We apologize for this oversight.

A thorough revision of the language is needed to enhance the readability of the text.

Authors #6

We have thoroughly revised the manuscript according to the comments and suggestions provided. Please see the revised version of the manuscript.

---

## [Editor Report · Decision Letter 2]

5 Sep 2023

Roles of the interaction with children and families in mediating the association between digital health literacy and well-being of early childhood teachers in Portugal: a cross-sectional study

PONE-D-22-23884R2

Dear Dr. Rosario,

We’re pleased to inform you that your manuscript has been judged scientifically suitable for publication and will be formally accepted for publication once it meets all outstanding technical requirements.

Kind regards,

Ali Garavand

Academic Editor

PLOS ONE

---

## [Editor Report · Acceptance letter]

12 Sep 2023

PONE-D-22-23884R2 

Roles of the interaction with children and families in mediating the association between digital health literacy and well-being of early childhood teachers in Portugal: a cross-sectional study 

Dear Dr. Rosário:

I'm pleased to inform you that your manuscript has been deemed suitable for publication in PLOS ONE. Congratulations! Your manuscript is now with our production department. 

Kind regards, 

on behalf of

Dr. Ali Garavand 

Academic Editor

PLOS ONE